# Morphological, Phylogenetic and Toxinological Characterization of Potentially Harmful Algal Species from the Marine Coastal Waters of Buenos Aires Province (Argentina)

Jonás Adrián Tardivo Kubis [1,*,†], Francisco Rodríguez [2], Araceli E. Rossignoli [3], Pilar Riobó [4], Eugenia A. Sar [1,*,‡] and Inés Sunesen [1,*,†,‡]

1   División Ficología Dr. Sebastián Guarrera, FCNyM, Paseo del Bosque s/n, La Plata 1900, Argentina
2   Departamento de Microalgas Nocivas, Instituto Español de Oceanografía (IEO-CSIC), Subida a Radio Faro 50, 36390 Vigo, Spain
3   Procesos Oceanográficos Costeiros, Centro de Investigacións Mariñas (CIMA), Pedras de Corón s/n, 36620 Vilanova de Arousa, Spain
4   Departamento de Fotobiología y Toxinología de Fitoplancton, Instituto de Investigaciones Marinas, CSIC, Eduardo Cabello 6, 36208 Vigo, Spain
*   Correspondence: jtardivokubis@fcnym.unlp.edu.ar (J.A.T.K.); easar@fcnym.unlp.edu.ar (E.A.S.); isunesen@fcnym.unlp.edu.ar (I.S.)
†   These authors pertaining to UNLP.
‡   These authors pertaining to CONICET.

**Abstract:** In the framework of a monitoring program of harmful microalgae from the marine coastal waters of the Buenos Aires Province, seven strains were isolated and characterized by morphological and molecular analysis (LSU rDNA partial sequencing, D1–D3 regions). Established strains belonged to *Alexandrium catenella*, *Protoceratium reticulatum* and *Pseudo-nitzschia multiseries*. The toxinological profile of the target strains were determined by UHPLC-FLD equipment for paralytic shellfish toxins (PSTs) and LC-MS/MS for lipophilic (LSTs) and amnesic toxins (ASTs). The toxin profile varied in the four strains of *A. catenella*, the predominant compounds were gonyautoxins (GTXs) GTX2,3 and GTX1,4 for strains LPCc001 and LPCc004, and N-sulfocarbamoyl toxins (Cs) C1,2 and GTX1,4 for strains LPCc002 and LPCc008. The obtained cellular toxicity values were moderate-to-high (12.38–46.40 pg saxitoxin equiv. cell$^{-1}$). The toxin profile of *P. reticulatum* was dominated by yessotoxins (YTXs) (up to 94.40 pg cell$^{-1}$) accompanied by homo-yessotoxin (Homo-YTX) traces. In *P. multiseries*, the toxin profiles were dominated by domoic acid (DA) (1.62 pg cell$^{-1}$ and 1.09 pg cell$^{-1}$) and secondarily by Isomer A (Iso-A), Epi-domoic acid (Epi-DA), Isomer-E (Iso-E) and Isomer-D (Iso-D). This study provides detailed information about representative HAB species in the area, useful for resource management, risk evaluation and related research on toxic dinoflagellates and diatoms.

**Keywords:** *Alexandrium catenella*; molecular analysis; morphology; *Protoceratium reticulatum*; *Pseudo-nitzschia multiseries*; toxinological profiles



## 1. Introduction

A phytoplankton and biotoxin monitoring program has been implemented since 2008 in marine coastal waters of the Buenos Aires Province to mitigate the impacts of harmful algal blooms (HABs) on the aquaculture and marine life, as well as to protect human health [1,2]. Some HABs can be responsible of high toxin levels in fishery products for human consumption. In the case of the Argentine Sea, the main toxins previously detected and reported in the literature are paralytic shellfish toxins (PSTs) and diarrhetic shellfish toxins (DSTs), both of which produced poisoning outbreaks in humans. Amnesic shellfish toxins (ASTs), yessotoxins (YTXs), azaspiracid shellfish toxins (AZTs) and spirolides (SPXs) have also been detected in seafood and phytoplankton from the area, but not related to

human intoxications [3,4] (and references therein). Typically, toxin-producing microalgae in the Argentine Sea belong to dinoflagellates from the genera *Alexandrium*, *Gymnodinium*, *Dinophysis*, *Prorocentrum*, *Protoceratium* and *Azadinium* as well as diatoms from the genus *Pseudo-nitzschia* [3] (and references therein).

In the framework of the monitoring program, several potentially toxigenic planktonic species associated with shellfish toxicity were recently isolated from that area: four clonal strains of the dinoflagellate *Alexandrium* sp., one of the dinoflagellate *Protoceratium* sp. and two of the diatom *Pseudo-nitzschia* sp. Species of these three genera were recorded as toxin producers in the Argentinean Sea, even with moderate cellular concentrations that do not cause water discolorations.

The genus *Alexandrium* Halim is globally distributed in coastal waters and includes about 33 currently accepted species [5], some of which are well known as PST producers and responsible for outbreaks of paralytic shellfish poisoning (PSP) [6]. Human illness from consumption of contaminated shellfish or fish has been profusely documented [6,7] (and references therein). Regarding Argentina, the first HAB documented occurred in 1980 associated with a PSTs producer, *Alexandrium catenella* (Whedon and Kofoid) Balech (reported as *Gonyaulax excavata* (Braarud) Balech). The outbreak caused human intoxication and the death of two fishermen in shelf waters off the Valdés Peninsula (Chubut Province, Argentina) [8,9]. Up to this date, thirty-two cases of paralytic shellfish poisoning (PSP) and seven human fatalities associated with *A. catenella* (reported as *A. tamarense*) have been formally registered in the Argentinean Sea [4] (Table 1, and references therein). Other species of PST producers from the genus *Alexandrium* found in the Argentinean Sea are *A. minutum* Halim (analysed from net tow samples, NT) [10] and *A. ostenfeldii* (Paulsen) Balech and Tangen (analysed from cultures) [11]. The latter species was also reported as SPX producing [10–12].

The genus *Protoceratium* Bergh emend. H.Gu and Mertens occurs worldwide in coastal waters and includes 10 currently accepted species [5], of which only *P. reticulatum* (Claparède and Lachmann) Bütschli being reported as YTX producing [13]. No human intoxications have been reported to date caused by YTXs [4,14]. The first report about YTXs in Argentinean shellfish [15] was based on the analyses of diverse shellfish species harvested over 20 years, from varying geographical sources. Akselman et al. [16] detected YTXs for the first time in phytoplankton samples from the SW Atlantic and in two strains of *P. reticulatum* (A1 and H1) isolated from the San Jorge Gulf. Afterwards, Fabro et al. [17] reported the presence of YTXs in phytoplankton field samples and identified *P. reticulatum* as the associated species.

Diatoms from the genus *Pseudo-nitzschia* Peragallo occur worldwide in coastal waters and include about 58 species, of which 27 are known to produce domoic acid (DA), a neurotoxin responsible for amnesic shellfish poisoning (ASP) or DA poisoning [18–20] (and reference therein). Nevertheless, since the first documented case of a hundred ill people and three human fatalities caused by DA associated with HAB of *P. multiseries* (Hasle) Hasle (as *Nitzschia pungens* f. *multiseries* Hasle) [21,22], no other human fatalities due to DA have been reported [19].

In the Argentine Sea, there have been nine reported toxigenic *Pseudo-nitzschia* species. Among these, only *P. australis* Frenguelli, *P. fraudulenta* (Cleve) Hasle, *P. multiseries* and *P. pungens* (Grunow ex Cleve) Hasle [4] (and references therein) have been demonstrated to produce DA. The first detection of DA in mussels (*Mytilus edulis*), anchovy viscera (*Engraulis anchoita*) and plankton samples from Mar del Plata (Buenos Aires Province), was associated with *P. australis* [23,24]. Instead, the first detection of DA in field samples from coastal waters of Chubut was associated with *P. fraudulenta* [25]. DA was also detected in a strain of *P. multiseries* isolated from shelf waters in the Buenos Aires Province [26]. Formerly, this species had been associated with the first detection of DA in mussels from Punta del Este, Uruguay [27].

The aim of this study was to provide an integrated morphological, molecular (LSU rDNA sequencing) and toxinological analysis of the target organisms isolated from Buenos

Aires marine, coastal waters and to compare these results with previous knowledge on cultures and field populations from Argentina and other geographical regions. The full characterisation of toxic microalgae associated with PST, YTX and AST production in the Argentine Sea being especially useful in the framework of the monitoring program to provide basic knowledge about profiles and content toxin useful for management purposes and mitigation strategies in the studied area.

## 2. Materials and Methods

### 2.1. Isolation and Established Cultures

Live samples were collected with 30 μm mesh net hauls along the marine coastal waters in Buenos Aires Province, Argentina (Figure 1).

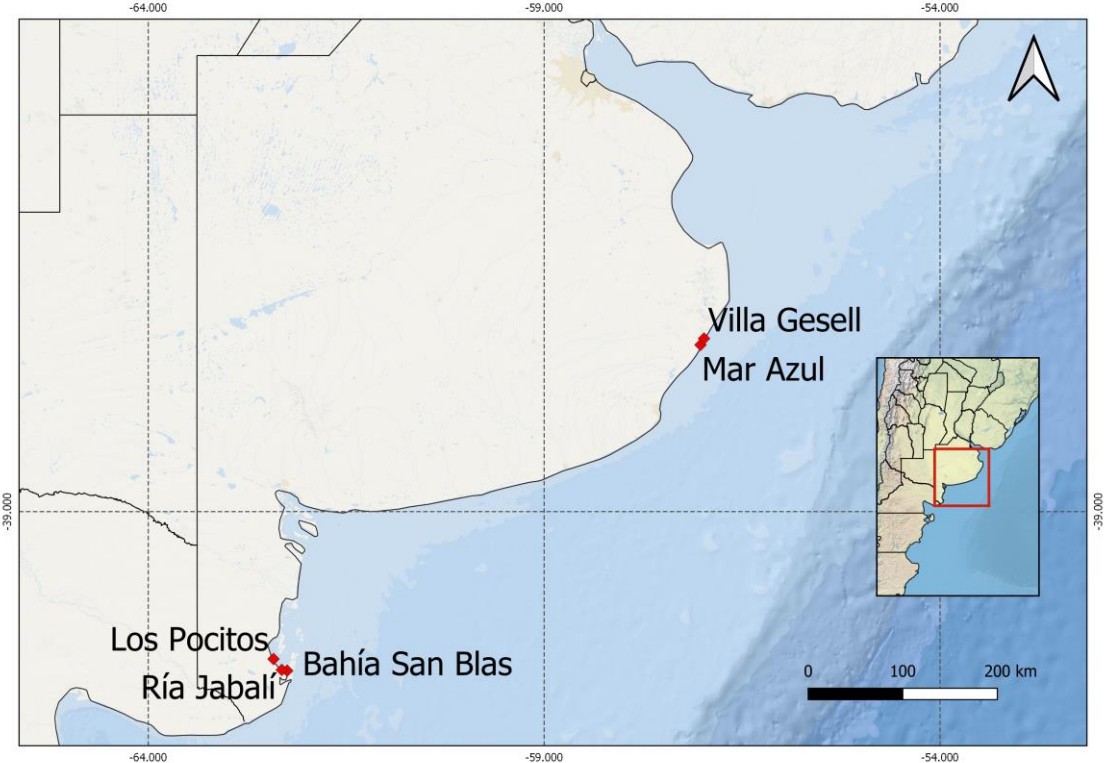

**Figure 1.** Map of sampling areas and location in Argentina.

Single cells were isolated by micropipette using a Zeiss Axiovert 40 CFL inverted microscope with phase contrast and differential interference contrast (DIC) (Zeiss Microimaging, Goettingen, Germany). Individual cells were washed several times in local filtered seawater and when free of contaminants they were transferred into 6-well tissue culture plates containing 10 mL natural seawater enriched with Guillard's f/2 medium without silicates for dinoflagellates (Sigma-Aldrich, Saint Louis, MO, USA) and with Guillard's f/2 medium with silicates for diatoms (Sigma-Aldrich). Cells were incubated at 16 °C, at salinity of 30 and under light supplied by cool-white fluorescent tubes with irradiance of 100–125 μmol photons $m^{-2}$ $s^{-1}$ on a 12:12 light:dark regime, in a growth chamber (SEMEDIC I-290F, SEMEDIC SRL, CABA, Argentina). After successful isolation, cultures were inoculated to 40 mL medium in 100 mL flasks and incubated in the described conditions. A total of seven strains were obtained, five of them belonging to the dinoflagellate genera *Alexandrium* and *Protoceratium*, and the rest to diatoms from the genus *Pseudo-nitzschia*. Further morphological and molecular characterisation was performed as detailed below.

## 2.2. Microscopy

For light microscopy (LM) analyses, cells were observed alive or fixed in 4% formaldehyde, using a Leica DMLA microscope (Leica Microsystems, Wetzlar, Germany) equipped with DIC. For the analyses of the cell thecal plates arrangement, specimens were stained with calcofluor white (Fluorescent Brightener 28, Sigma, St. Louis, MO, USA) following Fritz, Triemer [28]. For the analyses of frustules of field material and strain cultures, samples were also treated with hydrogen peroxide (30% $w/v$—100 vol) at 90 °C for 4 h [29], to remove the organic matter. Photographs were taken with the digital camera AxioCam HRc (Carl Zeiss Microscopy GmbH, Jena, Germany). Cell counts were conducted using a Zeiss Axiovert 40 CFL inverted microscope (Carl Zeiss Microscopy GmbH, Jena, Germany) by Utermöhl technique [30].

For scanning electron microscopy (SEM) analyses, cells or treated frustules (in the case of diatom samples) were fixed in 4% formaldehyde final concentration. One mL of cells/frustules suspension was collected on nylon (polyamide) filter membranes (13 mm diameter, 0.45 μm pore size, Sartolon polyamide Sartorius Stedim Biotech, Goettingen, Germany) or PTFE (Teflon) filter membranes (13 mm diameter, 1 μm pore size, Gamafil S.A., Beccar, Argentina), in a filter funnel. Samples were rinsed in filtered seawater several times, and distilled water added in increasing proportions several times [31]. Filters were dehydrated in an ethanol series of 30%, 50%, 70%, 90%, 95% and 100% with 10 min at each concentration, and then critical-point dried (BalTec, model CPD-30, Balzers, Liechtenstein). Filters were mounted on aluminium stubs, sputter coated with gold with a JFC 1100 FC (JEOL, Tokyo, Japan) and subsequently observed with a JSM 6360 LV (JEOL, Tokyo, Japan) or, alternatively, with gold palladium with a Cressington 108 (Cressington Scientific Instruments, Watford, UK) and subsequently observed with an NTS SUPRA 40 FE-SEM (Carl Zeiss Microscopy GmbH, Oberkochen, Germany). Micrographs were analysed using Image J software [32].

## 2.3. Deposited Material

Field material, treated material, slides and SEM-stubs were deposited in the Herbarium of the División Ficología (LPC), Facultad de Ciencias Naturales y Museo, Universidad Nacional de La Plata, under collection numbers LPC11508, LPC11564, LPC12039, LPC12040, LPC12078 and LPC12080.

## 2.4. DNA Extraction, Amplification and Sequencing

An aliquot of 1.5 mL of late exponential growing cultures was concentrated by centrifugation, washed in two drops of milli-Q water, placed in 200 μL microtubes, cold shocked in liquid nitrogen and kept at −20 °C until further analysis. DNA extraction was performed using Chelex® (Bio-Rad, Hercules, CA, USA), following Richlen, Barber [33]. Then, liquid nitrogen cold shocking was also conducted, and samples kept at −20 °C before PCR analyses.

The domain D1–D3 of the LSU rDNA were amplified using the pairs of primers D1R/LSUB (5′–ACCCGCTGAATTTAAGCATA–3′/5′–ACGAACGATTTGCACGTCAG–3′; [34,35]) and the amplification reaction mixtures (20 μL) were performed using Horse-Power™ Taq DNA Polymerase MasterMix (Canvax, Spain) following manufacturer's instructions. The DNA was amplified in a Surecycler 8800 thermocycler (Agilent Technologies, Santa Clara, CA, USA) as follows: 4 min denaturing at 94 °C, followed by 30 cycles of 30 s denaturing at 94 °C, 1 min annealing at 54 °C and 2 min elongation at 72 °C, with an elongation step of 10 min at 72 °C. PCR reactions were verified by agarose gel electrophoresis (1% TAE, 80 V) and GelRed™ nucleic acid gel staining (Biotium, Hayward, CA, USA). PCR products were purified with ExoSAP-IT™ (USB Corporation, Cleveland, OH, USA) and purified DNA was sequenced using the Big Dye Terminator v3.1 Reaction Cycle Sequencing kit (Applied Biosystems, Foster City, CA, USA) and migrated in an AB 3130 sequencer (Applied Biosystems) at the CACTI sequencing facilities (University of Vigo, Vigo, Spain).

*2.5. Phylogenetic Analyses*

The obtained LSU rRNA gene sequences were inspected and aligned using MEGA X software [36]. Sequences from *Lingulodinium polyedra* (Stein) Dodge and *Gonyaulax spinifera* (Claparède and Lachmann) Diesing were used to root the LSU rDNA tree for *Alexandrium* and *Protoceratium*, respectively, and *Fragilariopsis* clade was used to root the LSU rDNA tree for *Pseudo-nitzschia*. The original alignments for the LSU rDNA phylogenies (including gaps) consisted of 770 bp for the *Alexandrium*, 921 bp for *Protoceratium* and 780 bp for *Pseudo-nitzschia*. The best evolutionary models for maximum likelihood (ML) phylogenetic analyses were estimated using the model selection tool in MEGA X software. The data file was tested for goodness of fit to 24 different nucleotide substitution models of evolution, and those with the smallest Bayesian information criterion (BIC) score were selected in each case.

Phylogenetic analyses of the studied organisms followed maximum likelihood (ML) and Bayesian inference (BI) methods, as detailed below.

For the *Alexandrium* phylogeny (44 sequences), ML analyses involved Tamura-Nei model [37] with invariable positions and Gamma distribution (5 categories (+G, parameter = 1.3121)). For the *Protoceratium* phylogeny (28 sequences), ML analyses involved Tamura-Nei model [37] with invariable positions and Gamma distribution (5 categories, +G, parameter = 1.2318). Finally, ML analyses for the *Pseudo-nitzschia* phylogeny (28 sequences) involved Kimura 2-parameter method [38] with invariable positions and Gamma distribution (5 categories, +G, parameter = 0.0500).

Bayesian inference (BI) analyses were carried out by sampling across the entire GTR model space using Mr. Bayes v3.2 [39]. The program parameters were statefreqpr, dirichlet (1,1,1,1); nst, mixed and rates, gamma. Phylogenetic analyses involved two parallel analyses, each with four chains. Starting trees for each chain were selected randomly using the default values for the Mr. Bayes program. The number of generations used in these analyses was 1,000,000. Posterior probabilities were calculated from every 100th tree sampled after log–likelihood stabilisation (burn-in phase). All final split frequencies were <0.02. The two methods rendered similar topologies and phylogenetic trees were elaborated using BI method, with bootstrap values and posterior probabilities from ML/BI analyses. The trees are drawn to scale, with branch lengths measured in the number of substitutions per site.

Net mean p-distances between clades of *Alexandrium* sp., *Protoceratium* sp. and the other clades, as well as between *Pseudo-nitzschia* sp. and the other clades were calculated using MEGA X. Thus, no corrections for multiple substitutions at the same site, substitution rate biases (e.g., differences in the transitional and transversional rates) or differences in evolutionary rates among sites were considered [40].

*2.6. Toxin Extraction and Analysis*

2.6.1. Paralytic Shellfish Toxins (PSTs)

For the strains of *Alexandrium* sp., LPCc001, LPCc002, LPc004 and LPCc008 Lugol-fixed aliquots of 5 mL were collected from cultures to determine cell density by LM using a Sedgewick-Rafter chamber. Cultures were harvested for toxin analysis during the mid-exponential growth phase and were filtered through 25 mm diameter glass fibre filters (Whatman). Each filter, containing the *Alexandrium* sp. cells, was placed in an Eppendorf tube. Then, 750 μL of 0.05 M acetic acid was added and the tubes were frozen at −20 °C until further use. Just before analysis, the contents of each tube were sonicated for 1 min at 50 watts (4710 Series Ultrasonic Homogenizer, 40 footswitch) and centrifuged at $5200 \times g$ and 10 °C for 10 min. The supernatant was collected in a clean Eppendorf tube. The extraction was repeated with another 750 μL 0.05 M acetic acid. Both supernatants were combined (final volume 1500 μL) and then filtered through 0.22 μm PTFE syringe filters prior to HPLC analyses.

The characterisation of PSTs in the dinoflagellate strains was carried out by ultra-performance liquid chromatography with fluorescence detection and post-column oxidation (UPLC-FLD-PCOX) method [41] with some modifications described by Salgado et al. [42]. The identification of N-sulfocarbamoyl toxins (Cs), C1 and C2 toxins, was achieved through the gonyautoxins GTX2 and GTX3, respectively, since acid hydrolysis of the sulphate group of the Cs give rise to the corresponding GTXs. The used certified reference PST standards were purchased from the National Research Council Canada (NRC-CRMs). The chromatographic LC column was a Waters XBridge® Shield RP, 4.6 × 150 mm, 3.5 µm and the injection volume was 20 µL. In order to determine the PST concentration in the samples, the external standard calibration procedure was used. According to the European Food Safety Authority [43], saxitoxin (STX) and toxicity equivalency factors (TEFs) were used to calculate the toxicity contribution of each toxin which was expressed as pg STX equivalents cell$^{-1}$. LOD (s/n = 3) and LOQ (s/n = 10) were 0.152 and 0.496 µg mL$^{-1}$ for GTX4; 0.114 and 0.256 µg mL$^{-1}$ for GTX1; 0.052 and 0.208 µg mL$^{-1}$ for GTX3 and 0.047 and 0.139 µg mL$^{-1}$ for GTX2, respectively. Sample toxin content was expressed as mol%, pg cell$^{-1}$ and fmol cell$^{-1}$.

### 2.6.2. Yessotoxins (YTXs) and Isomers

For the *Protoceratium* sp. strain, LPCc021, Lugol-fixed aliquots of 1 mL were collected from cultures to determine cell density by LM using a Sedgewick Rafter chamber. Lipophilic toxins were extracted by centrifuging the 58 mL aliquots of cultures that were harvested for toxin analysis during the mid-exponential growth phase at 2600 g for 10 min at 4 °C. The cell-free supernatants were carefully discarded and 1500 µL of MeOH 100% was added to the pelleted cells. These suspensions were transferred to 2 mL conical microcentrifuge tubes fitted with screw caps (Thermo Scientific, Waltham, MA, USA) and containing 0.5 mm glass beads (Soda Lime, Bio Spec Products, Inc., Bartlesville, OK, USA). Vigorous shaking of the samples for 20 s using a bead beater disrupted the cell membranes and released the cytoplasmic contents into the methanol. Light microscopy observation was used to confirm that all cells were broken. Finally, the flask content was filtered through 0.22 µm PTFE syringe filters prior to LC-MS/MS analyses.

YTXs in strain LPCc021 were carried out on an Exion LC AD™ System (SCIEX, Framingham, MA, USA) coupled to a Qtrap 6500 + mass spectrometer (SCIEX) through an IonDrive Turbo V interface in electrospray mode according to Rossignoli et al. [44]. Briefly, the toxins were separated in a Phenomenex Kinetex EVO C18 "core-shell" column 50 mm (length) × 2.1 mm (id), 2.6 µm (particle size). Mobile phase A was water and B MeCN 90%, both containing 6.7 mM NH4OH (pH 11). The gradient started with 22% B, was maintained for 0.1 min, followed by a linear increment to reach 95% B at minute 1.8 and maintaining this composition until minute 2.9. The composition was then returned linearly to the initial one in 0.2 min and maintained for 0.5 min before the next injection. The flow rate was 1000 µL min$^{-1}$, the injection volume was 1 µL and the column temperature was 40 °C. The used certified reference YTX standards were obtained from CIFGA, S.A. (Lugo, Spain). LOD (s/n = 3) and LOQ (s/n = 10) were 0.002 and 0.007 µg mL$^{-1}$ for YTX and 0.0007 and 0.002 µg mL$^{-1}$ for Homo YTX, respectively.

The mass spectrometer parameters were set to Ion source Gas 1, 75 (arbitrary units); Ion source Gas 2, 75 (arbitrary units); Ion spray voltage, 5000 (positive) and −4500 (negative); Capillary temperature, 600 (°C); Curtain gas, 30; Collision Gas, medium. Specific MS/MS fragmentation conditions and collision energies for YTXs are shown in Table 1.

**Table 1.** MS/MS fragmentation conditions for yessotoxin (YTX) determination.

| Toxin | ESI | Q1 | Q3 | CE (v) |
|---|---|---|---|---|
| YTX | NEG | 570.43 | 467.40 | −42 |
| YTX | NEG | 570.43 | 396.40 | −42 |
| Homo–YTX | NEG | 577.40 | 474.40 | −42 |
| Homo–YTX | NEG | 577.40 | 403.40 | −42 |

ESI = Electrospray ionisation mode, Q1 = m/z ratio in the first quadrupole, Q3 = m/z ratio in the third quadrupole and CE (v) = collision energy.

### 2.6.3. Domoic Acid (DA) and Isomers

For the *Pseudo-nitzschia* sp. strains, LPCc036 and LPCc037, Lugol-fixed aliquots of 1 mL were collected from cultures to determine cell density by LM using a Sedgewick Rafter chamber. Cultures were harvested for toxin analysis during the mid-exponential growth phase and were filtered through 25 mm diameter glass fibre filters (Whatman). Each filter, containing *Pseudo-nitzschia* sp. cells, was placed in an Eppendorf tube. Then, 750 μL of MeOH/water (50/50, *v/v*) was added. Before analysis, the contents of each tube were sonicated for 1 min at 50 watts (4710 Series Ultrasonic Homogenizer, 40 footswitch) and centrifuged at 10,395× *g* and 10 °C for 10 min. The supernatant was collected in a clean Eppendorf tube. The extraction was repeated with another 750 μL of MeOH/water (50/50, *v/v*). Both supernatants were combined (final volume 1500 μL) and frozen to −20 °C. Then, they were filtered through 0.22 μm PTFE syringe filters prior to HPLC analyses.

DA was analysed in strain LPCc036 and LPCc037 by LC/ESI-MS/MS using an Accela UHPLC system coupled to a triple quadrupole mass spectrometer TSQ Quantum Access MAX (Thermo Fisher Scientific, San Jose, CA, USA) equipped with a heated electrospray ionisation source HESI-II. The toxins were separated using a Kinetex C18 (50 × 2.1 mm, 2.6 μm) reversed–phase chromatographic column (Phenomenex, Torrance, CA, USA) maintained at 35 °C. Mobile phases A and B were, respectively, 0.2% *v/v* formic acid in water and 0.2% *v/v* formic acid in 50% *v/v* methanol (pH 2.4) [45]. The gradient started at 100% A and maintained at 100% until minute 2; then, the percentage of this phase was linearly reduced to 45% in minute 4 and maintained in those conditions for 2 min. Finally, the conditions were reverted to the initial ones in 1.4 min in order to equilibrate the column. Total run time was 9 min. Flow rate was 280 μL min$^{-1}$ and the injection volume was 5 μL. The ion transfer tube temperature and the HESI-II vaporiser temperature were set at 250 °C and 100 °C, respectively. Nitrogen was used as sheath and auxiliary gas at 20 and 10 arbitrary gas pressure units, respectively. The mass spectrometer was operated in positive ionisation mode with a spray voltage of 3500 V. Detection was carried out in the SRM mode using argon (>99.999%) as CID gas at a pressure of 1.5 mTorr. The transition 312.1 > 266.1 (collision energy = 15 V) was used for quantification and 312.1 > 248.1 (collision energy = 17) for confirmation. The used reference DA standards were purchased from the National Research Council Canada (NRC–CRMs). LOD (s/n = 3) and LOQ (s/n = 10) of the method for DA are 0.0079 and 0.263 μg mL$^{-1}$, respectively.

Additionally, for the *Pseudo-nitzschia* sp. Strains, another analysis was run to verify the presence of DA isomers. For this purpose, equipment and conditions were changed. An Exion LC ADTM System (SCIEX, Framingham, Massachusetts, USA) coupled to a Qtrap 6500 + mass spectrometer (SCIEX) through an IonDrive Turbo V interface in electrospray mode was employed in this case. The toxins were separated using a Luna C18 150 × 2 mm, 5 μm column, maintained at 30 °C. Mobile phases were the same as before for the DA analysis [45]. The LC operation mode was isocratic with 70% A and 30% B at 0.18 mL min$^{-1}$. Total run time was 50 min. Multiple reaction monitoring (MRM) from m/z 312.1 to 266.0 followed by an enhanced production (EPI) of 312.1 from m/z 100 to 315 scan modes in positive ionisation with a collision energy of 40 v were employed for DA and isomers detection. The mass spectrometer parameters were set to Ion source Gas 1, 70 (arbitrary units); Ion source Gas 2, 70 (arbitrary units); ion spray voltage, 5000 (positive); Capillary

temperature, 600 (°C); Curtain gas, 30 and Collision gas, high. The identification of DA isomers was possible by comparing the elution times with their counterparts disposed of the reference standard used for DA quantification.

## 3. Results

Detailed information on the strains of *Alexandrium catenella*, *Protoceratium reticulatum* and *Pseudo-nitzschia multiseries* treated in this study are shown in Table 2. In addition, three strains from *Alexandrium affine* (H.Inoue and Y.Fukuyo) Balech, *Pseudo-nitzschia americana* (Hasle) Fryxell and *Pseudo-nitzschia pungens* (Grunow and P.T.Cleve) Hasle, which were only included in the phylogenetic analysis, are also presented in Table 2.

**Table 2.** Detail of strains of *Alexandrium catenella*, *Protoceratium reticulatum*, *Pseudo-nitzschia multiseries*, *Alexandrium affine* *, *Pseudo-nitzschia americana* * and *Pseudo-nitzschia pungens* * isolated from samples collected in Buenos Aires coastal waters, including place of sampling, date and Genbank accession number.

| Species | Strain Label | Collecting Site | Collecting Date | Gen Bank Acc. No |
|---|---|---|---|---|
| *Alexandrium catenella* | LPCc001 | Los Pocitos (40°25′53″ S; 62°25′5″ W) (LP) | 21 July 2015 | MZ838943 |
| *Alexandrium catenella* | LPCc002 | San Blas Bay (40°33′9″ S; 62°3′37″ W) (BSB) | 4 August 2016 | MZ838944 |
| *Alexandrium catenella* | LPCc004 | Los Pocitos | 27 September 2016 | not available |
| *Alexandrium catenella* | LPCc008 | Ria Jabalí (40°32′9″ S; 62°18′57″ W) (RJ) | 21 July 2015 | MZ838945 |
| *Protoceratium reticulatum* | LPCc021 | Villa Gesell (37°17′10″S; 56°59′12″ W) (VG) | 17 October 2017 | MZ838951 |
| *Pseudo-nitzschia multiseries* | LPCc036 | Mar Azul (37°20′38″ S; 57°1′31″ W) (MAZ) | 29 January 2019 | MZ838946 |
| *Pseudo-nitzschia multiseries* | LPCc037 | Mar Azul | 29 January 2019 | MZ838947 |
| *Alexandrium affine* * | LPCc012 | Mar Azul | 9 January 2017 | MZ838950 |
| *Pseudo-nitzschia americana* * | LPCc039 | Villa Gesell | 22 April 2019 | MZ838949 |
| *Pseudo-nitzschia pungens* * | LPCc038 | Villa Gesell | 22 April 2019 | MZ838948 |

* Only included in the molecular analyses.

### 3.1. Morphological Analysis

The description of the species has been made exhaustively in order to account for the morphological variability found and with the purpose of facilitating the comparison of our results with those of the literature.

### 3.1.1. *Alexandrium catenella* (Whedon and Kofoid) Balech (Figure 2A–N)

References: [46] (37, Figure 2a–c); [47] (48, pl. 10, Figures 1–31, pl. 11, Figures 1–12); [48] (92, Figure 6a–d, as *A. tamarense*); [10] (1210, Figure 1a–m); [11] (81, Figure 8a–i).

Single cells or two-celled chains (Figure 2A–C), with many golden-brown elongated chloroplasts and a horseshoe-shaped nucleus are located in the equatorial part of the cell (Figure 2B). Cells are from small to relatively large sized, 21–46 μm long and 20–46 μm wide, slightly dorsoventrally flattened (Figure 2J,K). The cell surface was smooth and ornamented with many scattered small pores (Figure 2H–L). Epitheca was conic convex and hypotheca asymmetrically trapezoidal, antapically flattened, almost equal in size (Figure 2A,B). Cingulum was descendent by about one cingular height, bordered by narrow lists. Sulcus was deep, bordered by narrow lists, broadened towards the antapical region (Figure 2J–L). The cells had the typical plate formula of the genus: Po, 4′, 6″, 6c, 5‴, 2⁗, and the sulcal plates were not completely discriminated, only the sulcal anterior plate (S.a.) (Figure 2F,M), the sulcal posterior plate (S.p.) (Figure 2J–L,N), the posterior left sulcal plate (S.s.p.) and posterior right sulcal plate (S.d.p.) were analysed (Figure 2K,L). The pore plate (Po) was irregularly rectangular–oval, longer than wide, with convex left margin and concave, convex or irregular right margin, and with oblique to almost straight dorsal and ventral margins (Figure 2G–I). The Po had a comma–shaped foramen with a large head and a thick terminal or lateral callus, accompanied by marginal pores, and lacked connecting pores in most of the observed specimens (Figure 2D,G–I). The first apical plate (1′) was rather narrow, variable in shape,

asymmetrically rhomboid with anterior and posterior truncated corners, and with direct connection with Po (Figure 2D,H,I). It had a ventral pore in the anterior, concave, right margin (Figure 2H) or in the limit between 1′ and 4′ with two conjugated notches: one in 1′ and one in 4′ (Figure 2I). The plate 2′ was the largest of the apical plates, hexagonal, limiting with the left side of Po (Figure 2H), while 3′ was the second largest plate, limiting with the dorsal margin of Po (Figure 2H,I) and 4′ was the smallest apical plate, limiting with the right margin of 1′ (Figure 2D,H). The precingular plates from 1″ to 4″ were more or less trapezoidal and 5″ and 6″ were pentagonal (Figure 2D,E,H,I). The smallest precingular plate was 6″, with a concave left margin (Figure 2D,E,H). The postcingular plates 1‴ to 3‴ are lower than 4‴ and 5‴ (Figure 2J–L). The right margin of 1‴ was limiting with the sulcus, as well as the antapical margin with the first antapical plate (1⁗) (Figure 2J–L). The antapical margins of 2‴ to 4‴ were limiting with both antapical plates 1⁗ and 2⁗ (Figure 2J–L). The antapical plate 2⁗ was transversally elongated (Figure 2J–L). The S.a. plate was arc–shaped, longer than wide, sometimes with a deep posterior sinus and a fold or plica obliquely descending from the left anterior margin towards the centre of the plate (Figure 2F,M). The S.p. plate was pentagonal, longer than wide to wider than long, asymmetric, with a small or large connecting pore, closer to the right margin, united to the edge by a groove, limiting with 5‴ (Figure 2J–L,N). Additionally, the posterior sulcal plate was limiting with the antapical margin of 4‴, 2⁗ and 1⁗ (Figure 2J–L). S.p. had an oblique anterior margin delimited by two conspicuous projections where a pair of sulcal plates, S.d.p. and S.s.p., fit (Figure 2K,L).

**Morphological remarks:** In culture, the species were only observed in solitary or in two–celled chains when dividing. There was a great variability in the morphometric data, with small to relatively large–sized cells, and in the morphology of the diagnostic plates: Po, 1′, S.a. and S.p., as shown in Figure 2. The Po (=APC) presented scattered marginal pores and rarely presented a connection pore, 1′ exhibited generally a ventral pore, the S.a. plate frequently exhibited a fold or plica and the S.p. plate was generally observed with a connection pore. These morphological attributes were confirmed in all the studied strains ($n = 153$).

**Taxonomical remarks:** *Alexandrium catenella* was reported as the causative organism of the first outbreak of PSP in Argentina described by Carreto et al. [9] under the name of *Gonyaulax excavata* (Braarud) Balech. *G. excavata* was later transferred to the genus *Alexandrium* Halim under the name of *A. excavatum* (Braarud) Balech and Tangen [49] and synonymised with *A. tamarense* (Lebour) Balech [47]. The revision of the genus *Alexandrium* (Halim) Balech based on phylogenetic analyses [50,51] made it possible to establish the existence of three well-supported species complexes. One of these was designated as the *A. tamarense* species complex. Subsequently, John et al. [52] split this species complex (based on mating incompatibilities, molecular, toxinological and morphological analyses) in five species, formerly known as Groups I–V: Group I, *A. fundyense*; Group II, *A. mediterraneum*; Group III, *A. tamarense*; Group IV, *A. pacificum*; Group V, *A. australiense*. The nomenclatural discussion about *A. fundyense* by [53] and [54] was solved by the Nomenclature Committee for Algae [55], and as a result *A. catenella* (Whedon and Kofoid) Balech stands as the valid name for Group I. The strain MDQ1096 isolated from Mar del Plata coastal waters [9], also analysed by Montoya et al. [56,57], and strain H-3-D10 isolated from the middle-shelf of the Argentine Sea (adjacent to the Buenos Aires Province) by Guinder et al. [11] were genetically characterised as *A. catenella* (former Group I) [11,58].

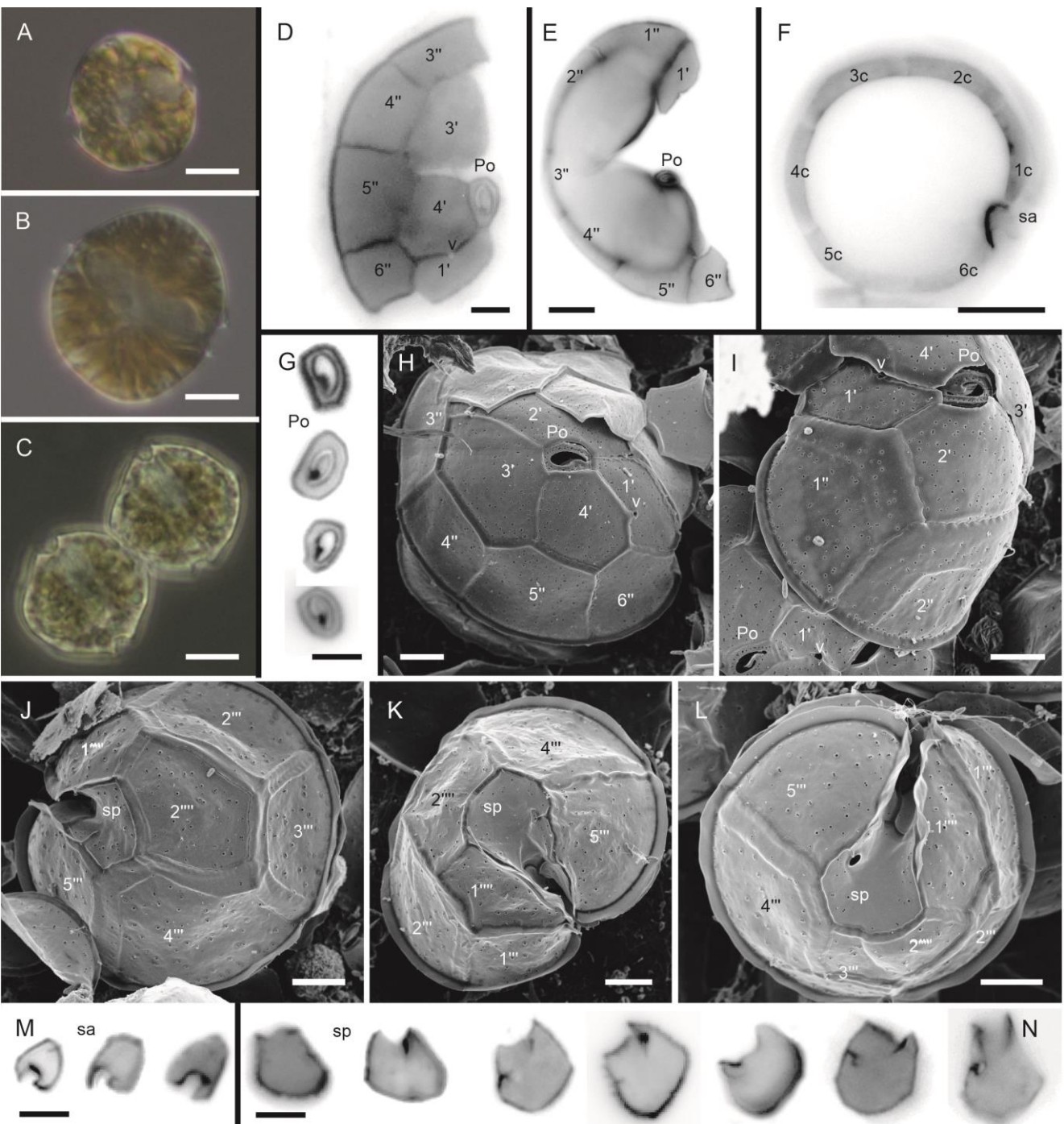

**Figure 2.** (**A–N**). *Alexandrium catenella*. Culture material. LM bright field, epifluorescence and SEM. (**A–C**). Cells of different sizes. Culture material. RJ 21-07-2015 (**A**), LP 27-09-2016 (**B**) and LP 21-07-2015 (**C**). (**D**). Detail of epitheca, Po and 1′ with ventral pore (v). (**E**). Open complete epitheca showing Po, 1′ with ventral pore (v) and six precingular plates. (**F**). Cingulum showing 6 cingular plates. Note the anterior sulcal plate (S.a.) between 1c and 6c. (**G**). Variations of the apical pore complex (Po). (**H,I**). Epitheca showing Po and 1′ with ventral pore (v). Note in Figure (**I**) the intercalary (growth) bands among plates. (**J–L**). Details of hypotheca. Note the posterior sulcal plate (S.p.) of different morphologies showing the connection pore. Note in (**J,L**) the intercalary (growth) bands. (**M**). Different sizes of anterior sulcal plate (S.a.). (**N**). Morphology and size variability of S.p. from culture material. Scale bar = 10 μm (**A–C,E,F**), 5 μm (**D,G–N**).

### 3.1.2. *Protoceratium reticulatum* (Claparède and Lachmann) Bütschli (Figure 3A–N)

References: [59] (169, pl. 77, Figures 1–5, as *Gonyaulax grindleyi*); [16] (44, Figure 2a–c); [17] (555, Figure 2a–I); [60] (6, Figures 5a–g, 6a–f and 7a–f).

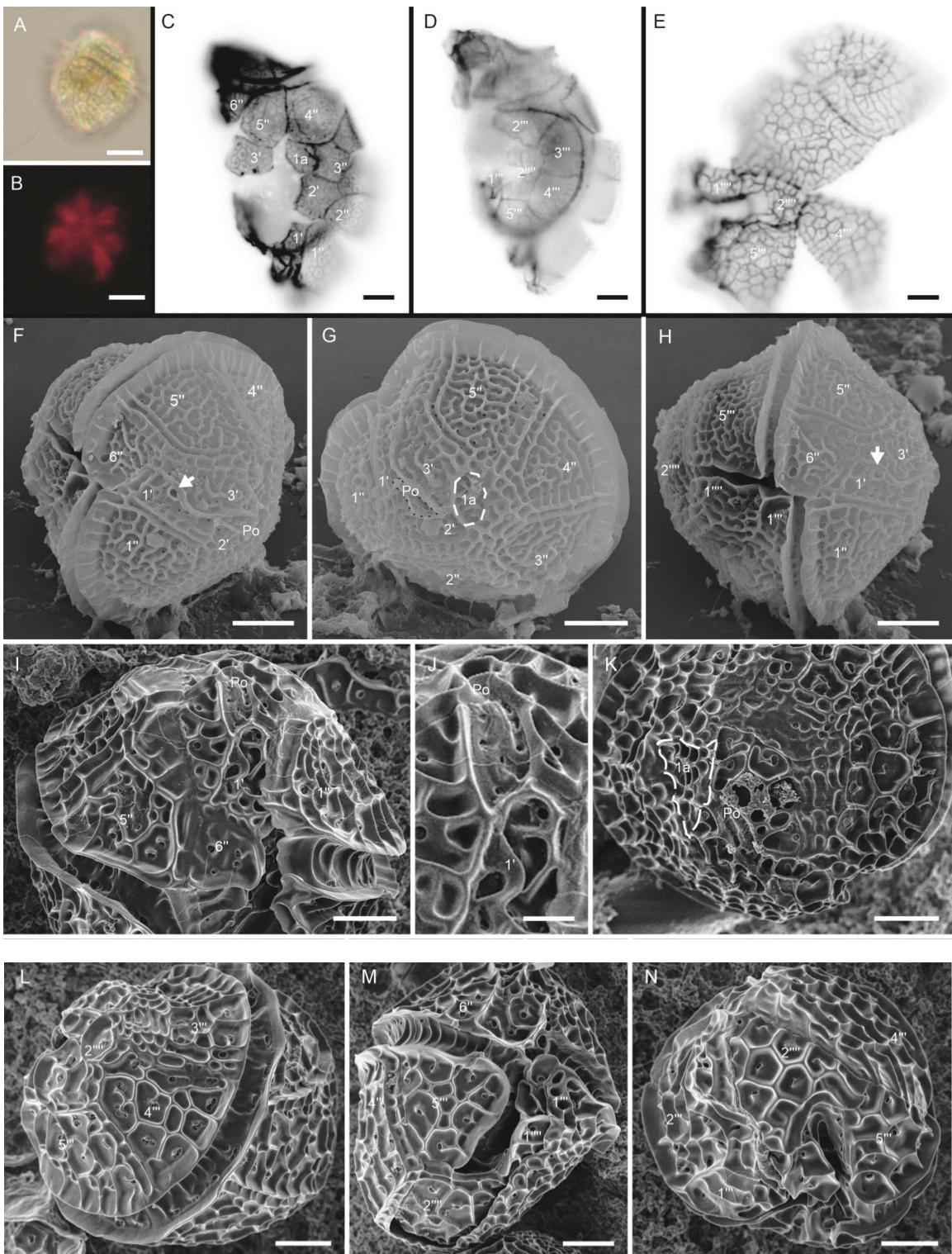

**Figure 3.** (**A–N**). *Protoceratium reticulatum.* Culture material. LPCc021 VG 17-10-2017 (**A–E,I–N**), Field material (**F–H**). LM Bright field, epifluorescence and SEM. (**A**). Live cell. (**B**). Chloroplast pattern. (**C,D**). Theca in different focuses, showing apical plates (**C**) and antapical plates (**D**). (**E**). Detail

of antapical plates. (**F–H**). Field material. Same specimen in different views. Figure 2F. Apical ventral view. Detail of epitheca showing apical pore complex (Po) and ventral pore (arrowed). Figure 2G. Apical view. Detail of plate 1a (encircled). Figure 2H. Ventral view. Note the position of antapical plates. (**I–K**). Epithecal plates. (**I**). View of Po, 1′ and 6″. (**J**). Details of apical pore complex (Po). (**K**). Apical view showing the 1a (encircled). (**L**). Cell in dorsal antapical view showing the trichocyst pores on the ornamented reticulated plates. (**M**). Cell in ventral antapical view. Note the 1‴″ and 2‴″. (**N**). Cell in antapical view. Scale bar = 10 μm (**A–H**), 5 μm (**I,K–N**), 2 μm (**J**).

Cells with numerous chloroplasts (Figure 3B) were variable in size, 22–45 μm long, 22–40 μm wide, 22–40 μm deep, subsphaeroidal- to polyhedral-shaped, with longitudinal axis larger than the transverse and epitheca slightly shorter than hypotheca (Figure 3A,H,M). The cell surface was reticulate, with ridges forming polygons and surrounding rimed pores or series of pores (Figure 3A,E–N). Cingulum was placed in the pre-equatorial part of the cell, descendent by about one cingular height, excavated, reticulate, bordered by narrow lists (Figure 3F,H,I,L,M). Sulcus was narrow and deep, bordered by narrow sulcal lists (Figure 3F,H,M,N). The cells had the typical plate formula of the genus: Po, 3′, 1a, 6″, 6c, 5‴, 2‴″, 6s. The pore plate (Po) was irregularly rectangular–shaped, with a large, elongated, central, slit–shaped pore surrounded by several rounded marginal pores (Figure 3F,G,I–K). The first apical plate (1′) was rhombic and had a conspicuous ventral pore in the right margin (Figure 3C,F,H,I). Plates 2′ and 3′ were larger and irregularly shaped (Figure 3C,G). The only intercalary plate (1a) was pentagonal and had a variable position, separated or in contact with the pore plate (Figure 3C,G,K). The smallest plate in the series of precingular plates was 6″. The first postcingular plate 1‴ was subtriangular and much smaller than the plates 2‴ to 5‴ of the series. The first antapical plate (1‴″) was smaller than 2‴″ and limiting with the left margin of the sulcus. The second antapical plate (2‴″) was located in the middle of the hypotheca and limiting with the sulcus as well as the postcingular plates 2‴, 3‴, 4‴ and 5‴.

**Morphological remarks:** The cells of *Protoceratium reticulatum* were variable in morphometric data, shape and morphology and position of the intercalary plate (1a) within the clonal strain (*n* = 60).

**Taxonomical remarks:** *Protoceratium reticulatum* was earlier recorded from Argentina as *Gonyaulax grindleyi* [59].

3.1.3. *Pseudo-nitzschia multiseries* (Hasle) Hasle (Figure 4A–I)

References: [61] (428, Figures 2, 4–5, 7, 8, 11–13, 17, 18); [62] (140, Figures 2, 7–9, 38–44); [63] (137, Figures 11–14); [64] (179, Figures 13–16).

The colonies were motile, stepped, formed by the overlapping of the tips of contiguous cells by one-third to one-fourth of the total cell length (Figure 4A). Cells were linear lanceolate to linear in valve and in girdle view (Figure 4A,B) with sharp ends and strongly eccentric, fibulate, raphe system, without central larger interspace (Figure 4B,D). Frustules were 68–130 μm long, 3.00–5.50 μm wide in field material, and 44–57 μm long, 2.58–4.18 μm wide, in cultured material. Valve surface was flattened, with interstriae alternating with striae undiscernible with LM and resolvable with SEM. The interstriae, internally elevated, had approximately the same density than fibulae, 12–16 in 10 μm and 13–17 in 10 μm, respectively (Figure 4D,F). The striae were multiseriate, with three to four rows of circular poroids (Figure 4D,F,H). The poroids, 5–8 in 1 μm, were circular, externally occluded by hymens (Figure 4H,I), and those placed close to the interstriae were slightly larger than the central (Figure 4D,F). Valve mantle was shallow, striated, striae of three to four poroid high (Figure 4C). The cingulum presented three bands, a valvocopula with 20–22 striae in 10 μm, 3–4 poroids high and 2–3 poroids wide, as well as two narrower copulae (Figure 4G,H).

**Morphological remarks:** The cultivated material presented a notable length reduction with respect to the field material, and the decrease became more drastic with the increase in the age of the culture (*n* = 90). The rest of the morphometric data of the isolated strains reasonably coincided with those of field material and those described for the species in the literature [63–65]. The changes in length were accompanied by alterations in the frustule outline in valve view. Several cells were either sickle-shaped (not shown), or lobed (Figure 4C,D,G,H). Lobes occurred along the margin of the theca/thecae in numbers of one (Figure 4H), two (Figure 4C,D,G) or three (not shown) near the central area of the frustule in the margin of the theca (Figure 4 C,D,G,H). The number of teratological cells in a culture increased with the increase in the age of the culture.

**Taxonomical remarks:** *Pseudo-nitzschia multiseries* was reported as the causative organism of the first outbreak of ASP in Canada, as *Nitzschia pungens* f. *multiseries* Hasle [21,22]. Subsequently, Hasle [66,67] reinstalled the genus *Pseudo-nitzschia* Peragallo, and established the combination *P. pungens* f. *multiseries* (Hasle) Hasle, and then [61] raised this form to the rank of species.

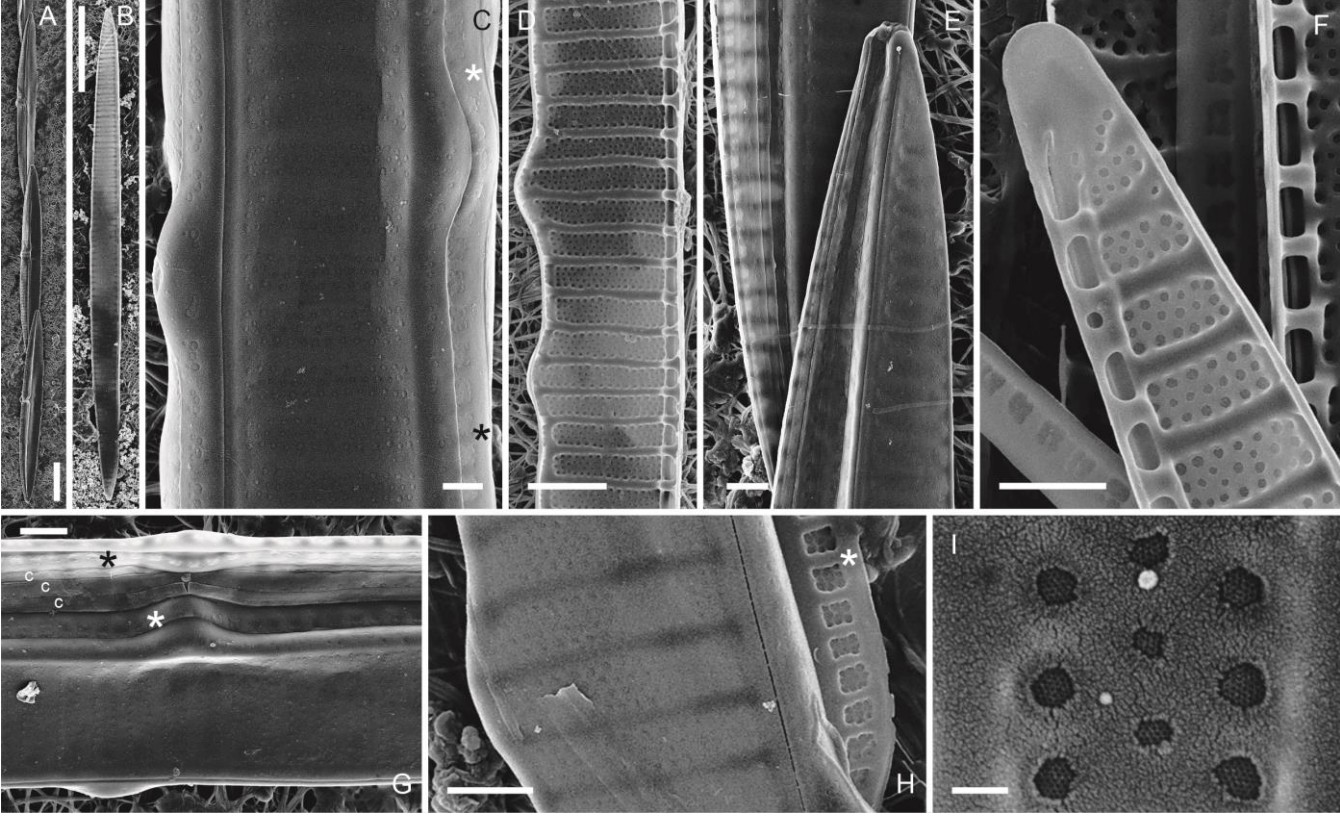

**Figure 4.** (**A–I**). *Pseudo-nitzschia multiseries*. SEM. Culture material LPCc036 MAZ 29-01-2019. (**A**). Chain of three cells. (**B**). Whole frustule in valve view. (**C**). External valve view of central part of the frustule showing the multiseriate striae of valve surface, mantle and valvocopula (*). Note two lobes along the theca margin. (**D**). Internal valve view of central part of valve. Note the lobes on one side of the valve. (**E**). External view of valve end. (**F**). Internal view of valve end. (**G**). External view of central part of the frustule showing valve, both valvocopulae (*) and copulae (c). Note lobes in the margin of two thecae. (**H**). Detail of external central part of valve striation and internal view of valvocopula (*). (**I**). Detail of cribra. Note a lobe along the theca margin. Scale bar = 10 μm (**A,B**), 2 μm (**D**), 1 μm (**C,E–H**), 100 nm (**I**).

### 3.2. Molecular Analysis

Phylogenetic analyses of the studied strains from the genera *Alexandrium*, *Protoceratium* and *Pseudo-nitzschia*, based on LSU rDNA D1–D3 regions, are shown in Figures 5–7, respectively.

The LSU phylogeny of *Alexandrium* (Figure 5) placed strains LPCc001, LPCc002 and LPCc008 in a well-supported clade belonging to *A. catenella* (ML: 100%), together with other species from the so-called *A. tamarense* complex. Accordingly, net mean distances (p) of the closest species to *A. catenella* clade in the LSU rDNA (D1–D3) were 0.082 for *A. tamarense*, 0.089 for *A. mediterraneum* U.John and 0.110 for *A. australiense* Sh.Murray.

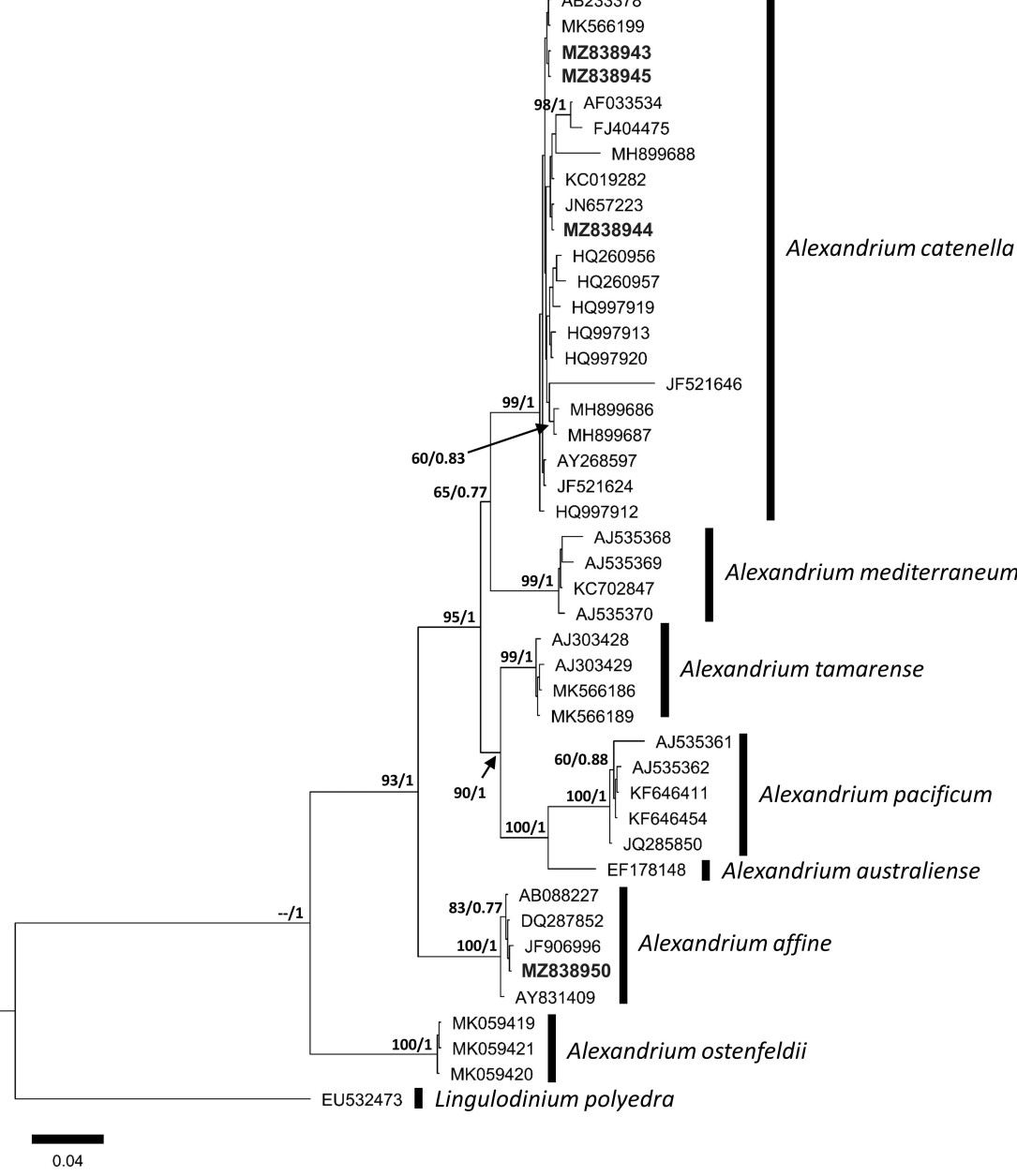

**Figure 5.** Phylogenetic tree of the D1–D3 LSU rDNA obtained by BI model showing the relationships among the *Alexandrium catenella* strains from Buenos Aires coastal waters and strains from other places in Argentina and around the world. Sequences of *A. ostenfeldii* from Argentina [11], and *A. affine* LPCc012, Genbank acc. number MZ838950 from Buenos Aires coastal waters. Numbers on branches are bootstrap percentages (*n* = 1000) and posterior probabilities (*n* = 1,000,000) after ML and BI analyses, respectively. Values lower than 60%/0.60 or not representative in one of the analyses are not shown or shown with hyphens, respectively.

Regarding the phylogenetic analysis of *Protoceratium reticulatum*, the strain LPCc021 belonged to a well-supported clade of *P. reticulatum* (ML 100%) (Figure 6), the sequences therein being identical. The closest species to *P. reticulatum* were *Pentaplacodinium saltonense* K.N.Mertens, M.C.Carbonell-Moore, V.Pospelova and M.J.Head (=*Ceratocorys mariaovidiorum* P. Salgado, S. Fraga, F. Rodríguez, P. Riobo and I. Bravo) (p-distance > 0.053), *Pentaplacodinium usupianum* Z.Luo, Leaw and H.Gu (p-distance > 0.055) and *Ceratocorys malayensis* Z.Luo, Leaw and H.Gu (p-distance > 0.096). Some differences in the topology of ML and BI trees were found regarding the relationships between *P. reticulatum*, *P. saltonense*, *P. usupianum* and *C. malayensis* due to the lower resolution observed in these cases in the ML phylogeny in comparison with the BI tree.

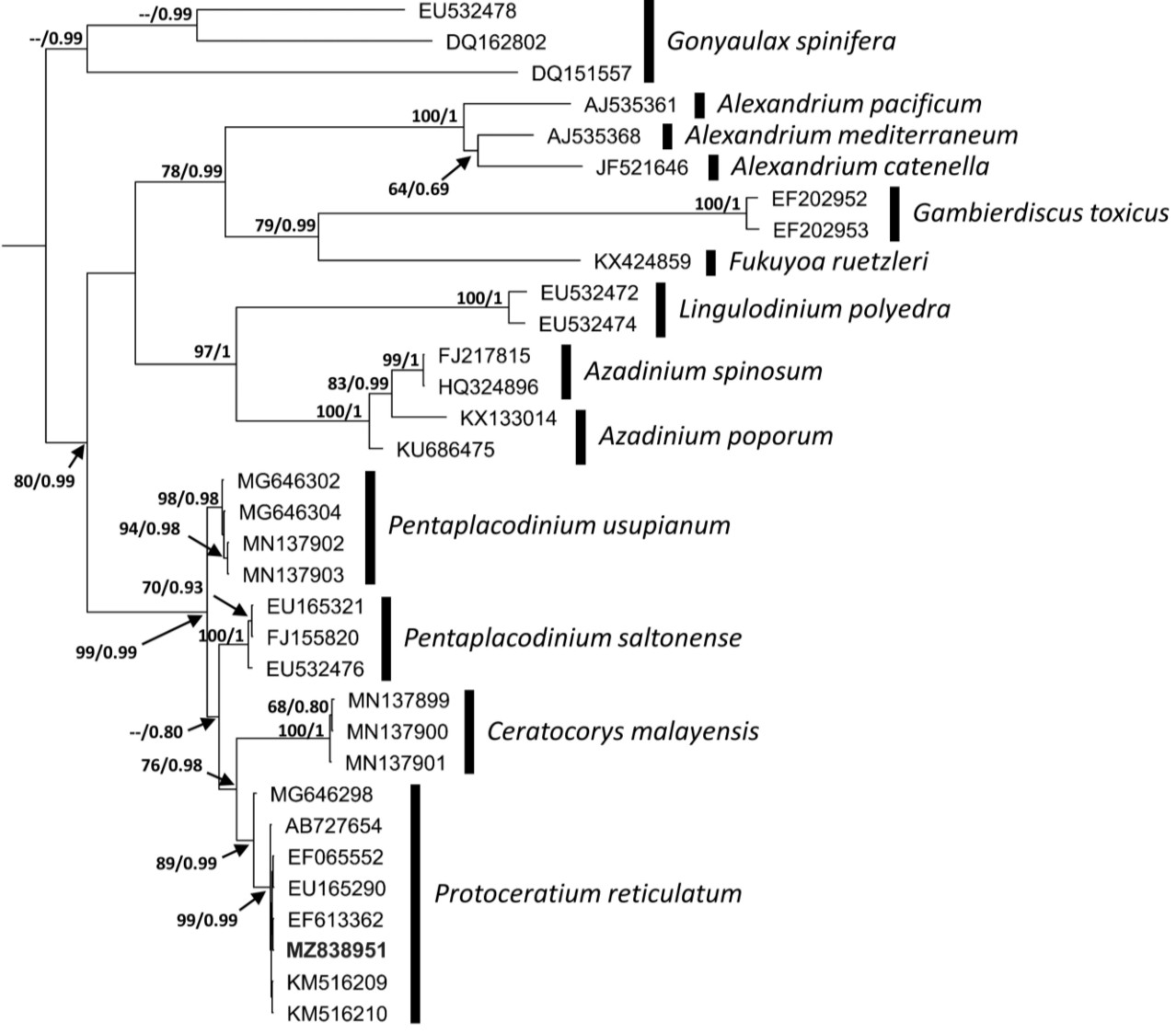

**Figure 6.** Phylogenetic tree of the D1–D3 LSU rDNA obtained by BI model showing the relationships among the *Protoceratium reticulatum* strains from Buenos Aires coastal waters and strains from other places in Argentina and around the world. Numbers on branches are bootstrap percentages (*n* = 1000) and posterior probabilities (*n* = 1,000,000) after ML and BI analyses, respectively. Values lower than 60%/0.60 or not representative in one of the analyses are not shown or shown with hyphens, respectively.

The LSU phylogeny of the genus *Pseudo-nitzschia* (Figure 7) placed the strains LPCc036 and LPCc037 in a well-supported clade of *P. multiseries* (ML: 100%) without any genetic differences among geographically distinct isolates. This species is closely related to the *P. pungens* clade, where LPCc038 was placed (ML: 96%). The net mean distances related to *P. multiseries* LPCc036 showed as closest species *P. pungens*, *P. australis*, *P. americana*, *Fragilariopsis* Hustedt and *P. brasiliana* Lundholm, Hasle and Fryxell (p-distance = 0.013, 0.019, 0.019, 0.020 and 0.022, respectively). Some differences in the topology of the ML and BI trees were found regarding the relationships between *P. multiseries*, *P. pungens*, *P. americana*, *P. brasiliana* and *P. australis*, associated with lower bootstrapping support in ML in comparison with the BI tree.

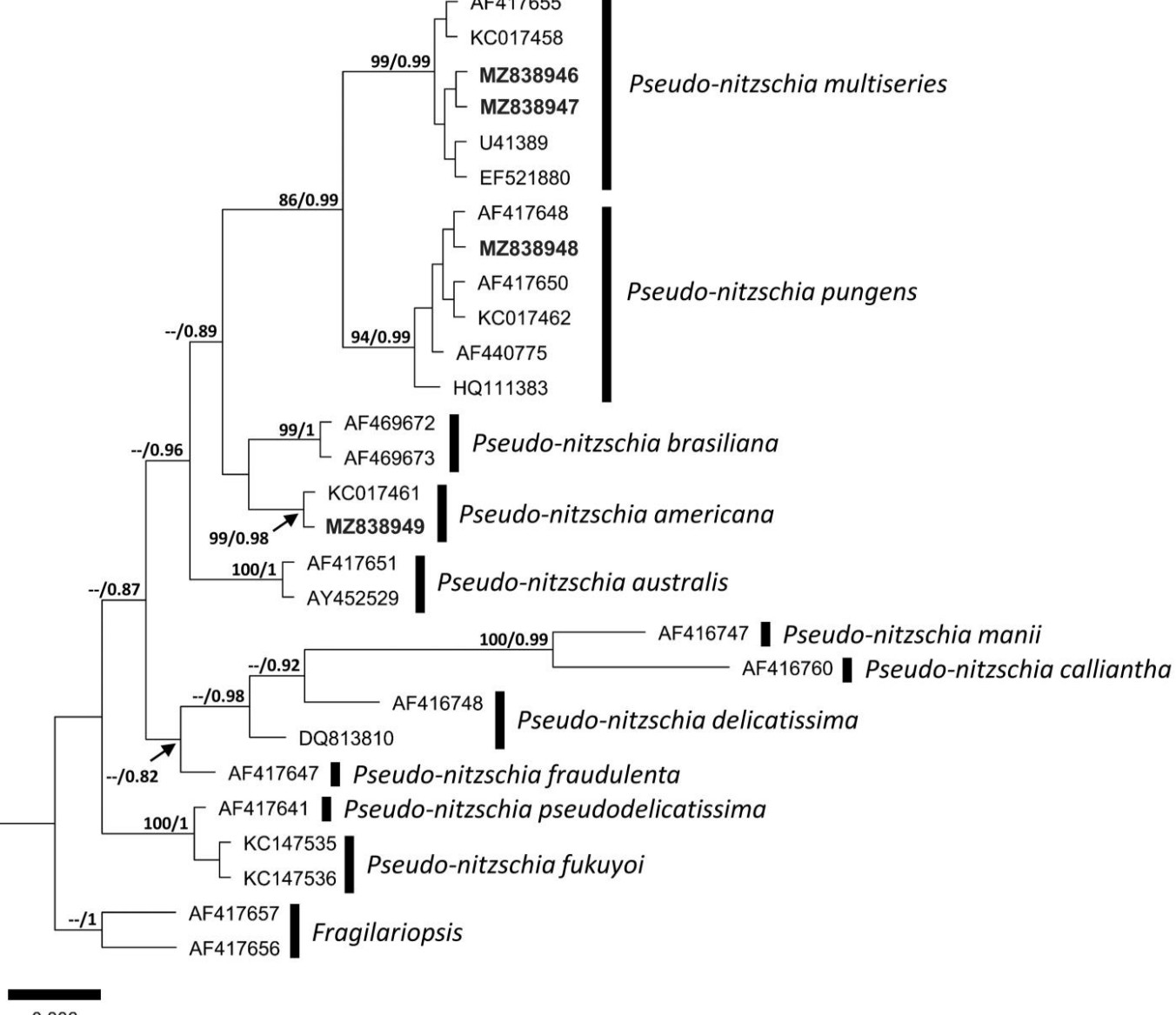

**Figure 7.** Phylogenetic tree of the D1–D3 LSU rDNA obtained by BI model showing the relationships among the *Pseudo-nitzschia multiseries* strains from Buenos Aires coastal waters and strains from other places in the world. The tree includes sequences of *P. americana* LPCc039, Genbank acc. number MZ838949, and *P. pungens* LPCc038, Genbank acc. number MZ838948, both from Buenos Aires coastal waters. Numbers on branches are bootstrap percentages (*n* = 1000) and posterior probabilities (*n* = 1,000,000) after ML and BI analyses, respectively. Values lower than 60%/0.60 or not representative in one of the analyses are not shown or shown with hyphens, respectively.

### 3.3. Toxinological Analysis

All the strains of *Alexandrium catenella* were found to contain STX-derived toxins. Toxicity values obtained were moderate-to-high (12.38–46.40 pg STX equiv. cell$^{-1}$) (Table 3).

**Table 3.** Toxin profile (total and isomers separately in brackets), toxicity and toxin content of *A. catenella* strains from Buenos Aires coastal waters. Gonyautoxins (GTXs), decarbamoyl toxins (dcGTXs), N-sulfocarbamoyl toxins (Cs).

| *A. catenella* Strains | Mol % | | | | Toxicity (pg STX eq. cell$^{-1}$) | Toxin Content (fmol cell$^{-1}$) | Toxin Content (pg cell$^{-1}$) |
|---|---|---|---|---|---|---|---|
| | GTX1,4 | GTX2,3 | dcGTX2,3 | C1,2 | | | |
| LPCc001 | 25.9 (18.5 + 7.4) | 33.7 (22.2 + 11.5) | 14.9 (12.2 + 2.7) | 25.6 (18.0 + 7.6) | 12.38 | 67.41 | 27.65 |
| LPCc002 | 20.2 (15.6 + 4.6) | 15.3 (10.8 + 4.5) | 18.3 (14.0 + 4.3) | 46.3 (32.8 + 13.5) | 27.59 | 194.45 | 81.57 |
| LPCc004 | 33.8 (22.1 + 11.7) | 37.6 (21.5 + 16.1) | 11.9 (9.9 + 2.0) | 16.8 (15.3 + 1.5) | 46.40 | 216.96 | 88.04 |
| LPCc008 | 30.9 (21.8 + 9.1) | 20.3 (14.7 + 5.6) | 13.5 (10.3 + 3.2) | 35.4 (24.8 + 10.6) | 46.09 | 255.85 | 106.53 |

The more toxic strains were LPCc004 (with 46.40 pg STX equiv. cell$^{-1}$) isolated from a sample collected in Los Pocitos in the early spring of 2016 and LPCc008 (46.09 pg STX equiv. cell$^{-1}$) isolated from a sample collected in Ría Jabalí in the winter of 2015. In opposition, the toxicity values from the strains LPCc001 isolated from a sample collected in Los Pocitos in the winter of 2015 and LPCc002 isolated from a sample collected in San Blas Bay in the winter of 2016 were the lowest (with 12.38 and 27.59 pg STX equiv. cell$^{-1}$, respectively) (Table 3). The analysed cultures produced moderate-to-high amounts of toxin content: 67.41–255.85 fmol cell$^{-1}$ and 27.65–106.53 pg cell$^{-1}$ (Table 3). Carbamate toxins (GTX4, GTX1, GTX3, GTX2), decarbamoyl toxins (dcGTX3, dcGTX2) and N-sulfocarbamoyl toxins (C1, C2) were the toxins detected for the isolates, while other PSTs were not detected or detected as trace amounts. The predominant toxins vary in the four strains of *Alexandrium*; for LPCc001 and LPCc004 the predominant toxins were the isomers GTX2, GTX 3 and GTX1, GTX 4, but in the strains LPCc002 and LPCc008, the predominant toxins were C1 and the isomers GTX1, GTX 4 (Table 3).

The strain of *Protoceratium reticulatum* LPCc021 showed the presence of the YTX (up to 94.40 pg cell$^{-1}$) and traces of Homo-YTX (<LOQ).

Finally, the strains of *Pseudo-nitzschia multiseries* LPCc036 and LPCc037 were found to contain DA (1.62 pg cell$^{-1}$ and 1.09 pg cell$^{-1}$, respectively). Isomer A (Iso-A), Epi-domoic acid (epi-DA), Isomer E (Iso-E) and Isomer D (Iso-D), from highest to lowest intensity signal, could also be identified in both strains by comparing with a reference CRM-DA standard (Figure 8). Two unknown peaks (?) with non-negligible intensities signals and not present in the standard, were also observed in samples after epi-DA elution (Figure 8). Their spectra are very similar to those of DA and specially Iso-A (Figure 9), so it is more than likely that they are other DA isomers that have not yet been identified.

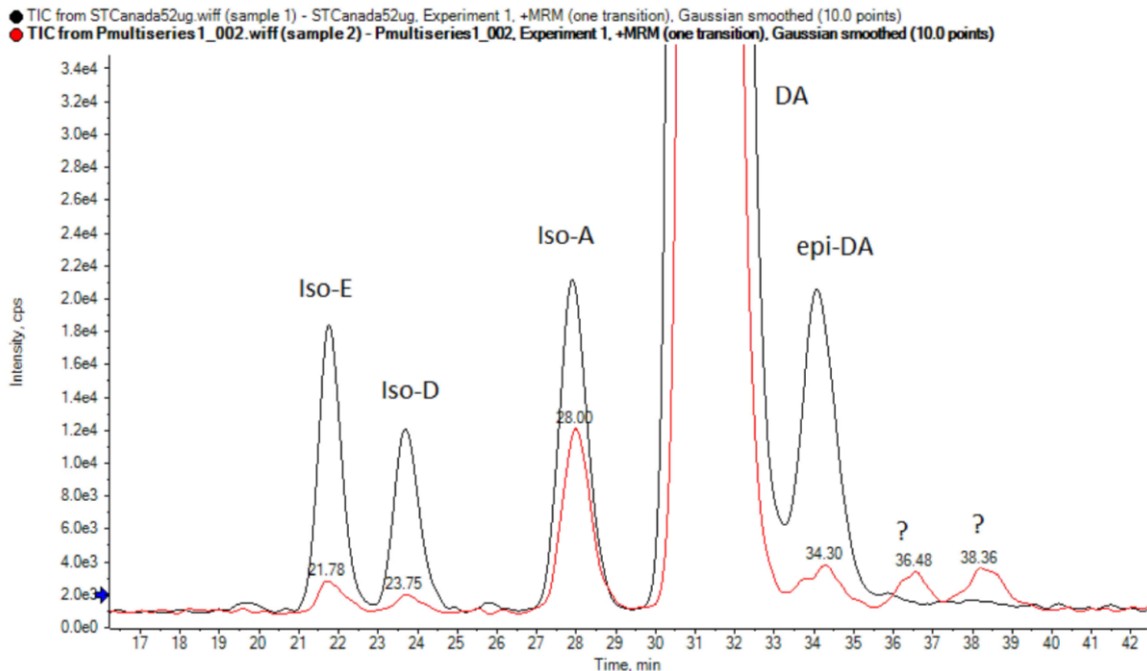

**Figure 8.** LC/ESI-MS/MS overlaid chromatograms of LPCc036 *Pseudo-nitzschia multiseries* strain (red line) and the reference CRM-DA standard (black line), showing the presence of DA and its isomers. ? unknown peaks.

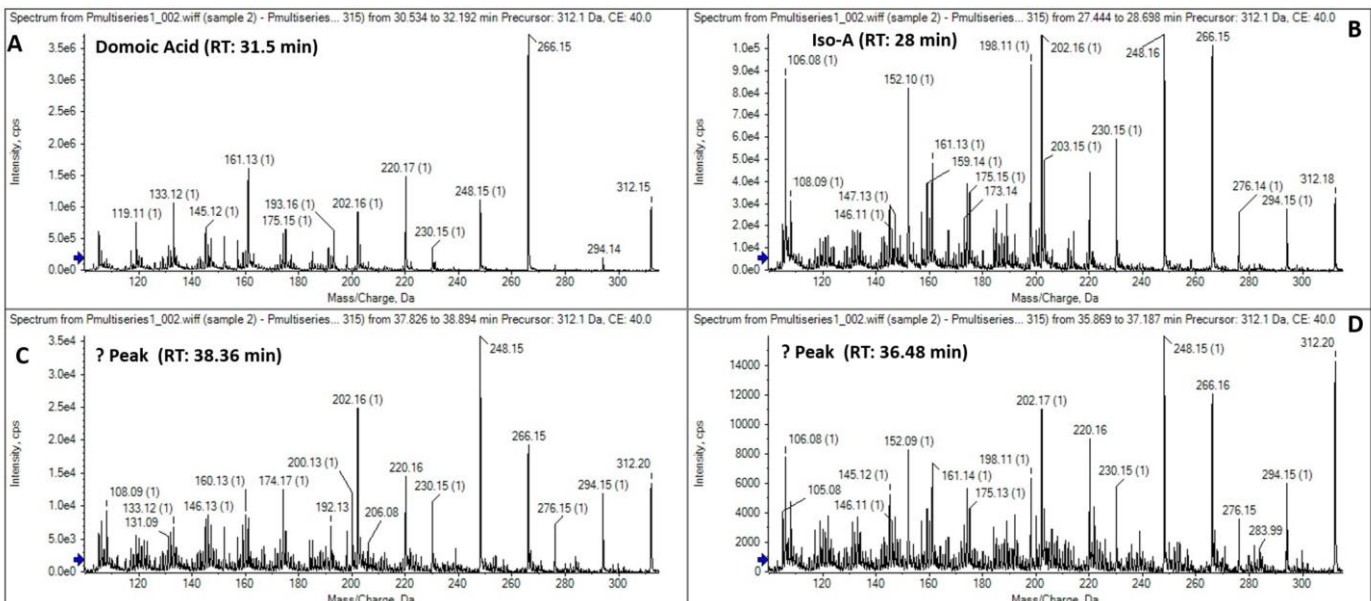

**Figure 9.** LC/ESI-MS/MS spectra peaks of LPCc036 *Pseudo-nitzschia multiseries* strain, corresponding to DA (**A**), Iso-A (**B**) and the two unidentified peaks (**C,D**). RT indicates the retention time at which the peak of interest elutes.

## 4. Discussion

### 4.1. Comparison of Morphometric and Morphological Features

The comparison between specimens of *Alexandrium catenella* analysed in this study and those described by Balech [47] made it possible to establish several differences in reference to specific morphological features. Some of these were similar to those described by Balech [47] (38, pl. 6, Figures 29–40) as differential characters to *A. tamarense*. The cells were

single or in couplets in the studied strains isolated from Buenos Aires coastal waters, while in Balech [47] the cells were forming curved chains in *A. catenella* and single in *A. tamarense*. According to MacKenzie et al. [68], *A. catenella* frequently forms long chains in the natural environment, and this character is markedly reduced in culture where cells appear single or in couplets. Field material from our study area never presents long chains. The cell size in the strains and field material analysed in this study (28–46 μm long, 24–46 μm wide) was somewhat larger than that described by Balech [47] (20.0–39.5 μm long, 22–44 μm wide), Fabro et al. [10] (25–37 μm long, 25–38 μm wide) and Guinder et al. [11] (26.4–42.9 μm long, 30–39.7 μm wide). The size range of *A. tamarense* given by Balech [47] (22–51 μm long, 22–50 μm wide) was broader than previously pointed out for *A. catenella*; however, this author commented that the smallest specimens were from Argentina.

The Po plate in the specimens analysed in this study had numerous marginal pores and generally lacked connecting pores, while the specimens described by Balech [47] had an elliptically shaped connecting pore near the comma head of the foramen in *A. catenella* and missing in *A. tamarense*. In reference to the connecting pore, it was consistently absent in the Po plate in both cultured and field material (this study, Figure 2D,G–I) and cultured material illustrated by Krock et al. [48] (Figure 6c,d, identified as *A. tamarense*), while it was present or absent in the strains illustrated by Guinder et al. [11] (Figure 8f,e, respectively) and present on a few cells of field specimens illustrated by Fabro et al. [10] (Figure 1c).

The first apical plate consistently had a small ventral pore on the right margin limiting with 4′ in cultured and field material in this study (Figure 2D,E,H,I), as well as in cultured material analysed by Krock et al. [48] (Figure 6c,d) and Guinder et al. [11] (Figure 8c–e, g), while it always lacked ventral pores in the specimens described by Balech [47]. *A. catenella* from the Bío-Bío region (Chile) [69] (Figure 2b,d–f,h) and from the China Sea [70] (Figure 1c,d,f) consistently lacked ventral pores in the 1′ plate found by Balech [47]. In the specimens of *A. tamarense* described in Balech [47], the ventral pore was always present, generally in 1′, in the limit between 1′ and 4′ with two conjugated notches: one in 1′ and the other in 4′ or in 4′, similar to what was also found by Fabro et al. [10] (Figure 1c,f,g,i) in field material of *A. catenella*.

The shape and size of the anterior and posterior sulcal plates were similar in specimens described in this study and those described by Balech [47] and Guinder et al. [11]; nevertheless, S.p. had a connecting pore in our material (Figure 2J–L,N) and Guinder's (Figure 8f,h), which was absent in Balech's. The morphological characteristics of our strains varied regarding those reported for the morphospecies examined by Scholin et al. [71], Lilly et al. [72], Penna et al. [58] and John et al. [52]. Based on the fact that morphological characters have shown a great diversity and a certain degree of variability [10,52,70], the determination of *A. catenella* was also confirmed by rDNA sequencing.

In the case of *Protoceratium reticulatum*, the size range of cultured and field specimens (22–45 μm long, 22–40 μm wide) was broader than those by Akselman et al. [16], from the Buenos Aires Province and San Jorge Gulf (34–39 μm long, 29–31 μm wide) and Balech [59], from the SW Atlantic Ocean (35–45 μm long, 28–37 μm wide, as *Gonyaulax grindleyi*). Instead, the size range of the specimens from platform and slope waters between 39 and 47° S (Argentine Sea) in Fabro et al. [17] were similar in amplitude (39.0–55.5 μm long, 33–48 μm wide) but larger in general than described in the present study.

*P. reticulatum* was described using different thecal formulas ([59]: Po, 3′, 1a, 6″, 6c, 6‴, 2⁗, 9s; [16]: Po, 3′, 1a, 6″, 6c, 6‴, 2⁗, ca 7s; [73]: Po, 4′, 0a, 6″, 6c, 5‴, 0p, 2⁗, 7s; [60]: Po, 3″, 1a, 6″, 6c, 5‴, 2⁗, 6s) creating confusion when comparing different materials due to interpretations about the designation of some plates. The most important difference in the epithecal formula is related to the relation between the plate 1a and the Po plate: if both plates were in contact, 1a was considered as an apical plate (3′) and the plate formula is 4′, 0a, 6″ [73]. Notwithstanding, considering that the position of 1a is variable even within the clonal strain, either separated from or contacting the pore plate, the epithecal formula is 3′, 1a, 6″ [16,59,60] and this study. In most of the cells analysed in the present work, the intercalary plate (1a) was separated from the Po plate, while in

Hansen et al.'s work [74] 50% of the cells showed the intercalary plate (1a) in contact with the Po. The most important difference in the hypothecal formula is related to the number of postcingular plates, 6''' [16,59] and 5''' [60,73]. In this study, we mentioned them as 1''' to 5''' as Salgado et al. [73], not *2''' to *6''' as Wang et al. [60]. Extra precingular plates previously recorded by Hansen et al. [74] in cultured material were also found in a cell in this study (8'' instead of 6'', Figure 3C). Based on the previous analyses, specimens in this study are morphologically similar with comparable variability to that described in the quoted literature.

Finally, regarding the studied strains of *Pseudo-nitzschia multiseries,* the reduction in cell length observed relative to the field material resembled that previously found in cultures from Monterey Bay California, USA, and the NW Sea of Japan [75,76]. Instead, the increase in original width described in cultures by Villac [75] was not observed in the strains from Buenos Aires. Moreover, the cell shape in our strains changed along the length reduction process, being more pronounced in old established cultures as reported by Subba Rao, Wohlgeschaffen [77], Villac [75] and Orlova et al. [76]. Nevertheless, the extreme alteration of the frustules and the formation of ribbon–shaped colonies instead of stepped chains described by Subba Rao, Wohlgeschaffen [77] (Figures 5–8 and 10, respectively), were not observed in our material.

### 4.2. Molecular Comparison

The molecular results in this study show that our three strains of *Alexandrium* corresponded to the ribo-species of *A. catenella* [52,54,55] with robust statistical support and a low intraspecific genetic distance. In agreement with other authors [10,11,58,78], this result confirms that strains of the *A. tamarense* species complex of Argentina always seem to belong to the ribo-species of *A. catenella*. Thus, both names *A. tamarense/A. catenella* can be found in local studies depending on the year of publication, but "true" *A. tamarense* (former Group III, [53]) does not produce PSTs and its type locality is the NE Atlantic (River Tamar, UK; [79]).

The genetic cluster of *A. catenella* included sequences from Argentinean, Chilean and Brazilian waters. These sequences are highly similar as shown in previous studies [52,58,80–82], so the hypothesis of a dynamic population in South America may be plausible [51,83,84].

The phylogenetic analysis of *Protoceratium reticulatum* reconstructed the evolutionary relationships between clades previously found by other authors. The BI tree of the LSU 28S rDNA of strains included in the multiple sequence alignment resembles those by Akselman et al. [16], Howard et al. [85] and Luo et al. [86], among closely related species and other toxin-producer dinoflagellates. Akselman et al. [16] suggested that within the clade of *Protoceratium* there were separated evolutionary units which they called *P. reticulatum* Clades A and B that could be different species morphologically coincident. Salgado et al. [87] analysed strains of *P. reticulatum* Clade B (including some previously considered by Akselman et al. [16]) and described it as *Ceratocorys mariaovidiorum* based on the consistent genetic differences and subtle morphological features regarding *P. reticulatum* Clade A. Subsequently, Mertens et al. [88] established that *C. mariovidiorum* was a junior heterotypic synonym of *Pentaplacodinium saltonense* gen. et sp. nov. [89], and Luo et al. [86] erected two species (*P. usupianum* and *C. malayensis*) genetically close to *P. reticulatum* Clade A. Our results (Figure 6) also corroborate the position of *P. saltonense*, *P. usupianum* and *C. malayensis* as close species to *Protoceratium reticulatum* [86,87,89].

The LSU phylogeny of *Pseudo-nitzschia* reconstructed the evolutionary relationships between clades previously reported by other studies [90–93]. The LSU clades displayed low genetic variability, supporting the identification of our strains.

### 4.3. Toxinological Comparison

The toxin composition of the studied strains of *Alexandrium catenella* showed variable toxin profiles and contents per cell. Previous toxinological studies [56,81–83,94] have also shown differences in the toxin profiles, contents per cell and toxicity. In addition, strains

acclimated under identical conditions (e.g., Varela et al. [81] and this study) also showed different toxin profiles and content per cell.

The strains LPCc001 and LPCc004 exhibited a toxin profile slightly dominated by GTX2,3 and secondarily by GTX1,4 (Table 3). This PST profile does not match those obtained in Argentina and Brazil [11,48,56,83,95], but it does with that of the strain SD01 from Santo Domingo (Aysén, Santo Domingo, Chile), dominated by GTX2,3 (4.7 GTX2 + 50.0 GTX3 mol% = 54.7 mol%) and secondarily by GTX1,4 (2.6 GTX1 + 29.6 GTX4 mol% = 32.2 mol%) [81].

The strains LPCc002 and LPCc008 presented a toxin profile dominated by C1,2 and secondarily GTX1,4 (Table 3) comparable to that described in other strains isolated from the Argentinean Sea [11,48,56,95] and southern Brazilian coastal waters [83] as *A. tamarense*. Similar toxin profiles (inversely dominated by GTX1,4 and secondarily by C1,2) were found in strains CB_Alex1 [82] and CB02 [81], from the Argentinean and Chilean sides of the Beagle Channel, respectively.

All our strains had a higher toxin content per cell and toxicity than the strain MDQ1096, from Mar del Plata shore (95, 63.20 fmol cell$^{-1}$ and 9.28 pg STX equiv. cell$^{-1}$) and those from Nuevo Gulf, San José Gulf and Valdés Peninsula (56, 13.30 to 35.42 fmol cell$^{-1}$ and a toxicity of 1.81 to 10.27 pg STX equiv. cell$^{-1}$). Additionally, with the exception of the strain LPCc001, all of our strains displayed higher toxin contents (Table 3) than the strains H5 (74 pg cell$^{-1}$) and H7 (60 pg cell$^{-1}$) from San Jorge Gulf, Argentina [48], and H-3-D10 (17.50 pg cell$^{-1}$) from Argentinean shelf adjacent to the Buenos Aires Province [11].

According to Carreto et al. [95] and Montoya et al. [56], a possible explanation for the variable toxin profiles and toxicity may be assayed culture conditions or environmental parameters (such as irradiance, temperature, salinity or inorganic nutrients).

The toxinological profile of *P. reticulatum* was dominated by YTXs, this result being characteristic for this species in Argentina. However, YTX concentration was much higher than those previously reported by Akselman et al. [16] in strains A1 and H1 (9.1 pg cell$^{-1}$ and 10.2 pg cell$^{-1}$, respectively) and Fabro et al. [17] from net hauls (2.2–12.5 pg cell$^{-1}$). Based on the comprehensive review by Paz et al. [14], who showed in their Table 1 that the highest concentration of YTX was determined in the strain 10628-OK-PR-C from Japan (59.8 pg cell$^{-1}$), the concentration found in our strain is the highest detected up to date. Additionally, the strain here analysed produces Homo-YTX traces (<LOQ), not detected previously by Akselman et al. [16] or Fabro et al. [17], but found by Ciminiello et al. [96] in strains from the Adriatic Sea (Italy), as a minor component of the YTX profile. Instead, Paz et al. [97] and Suzuki et al. [98] found it as the principal component of the YTX profile in a strain isolated from Vigo and Japan, respectively.

Regarding *Pseudo-nitzschia multiseries*, the strain isolated by Montoya et al. [26] from shelf waters of the Buenos Aires Province was mildly toxic (0.13 pg DA cell$^{-1}$), clearly below the range described in other geographical areas. In this sense, although our strains also displayed low DA amounts, these represented an order of magnitude higher than in Montoya et al.'s work [57], and fell within the range cited by former authors [99–103].

The toxin profile of our strains of *P. multiseries* was dominated by DA, secondarily accompanied from highest-to-lowest intensity signal by Iso-A, epi-DA, Iso-E, Iso-D and two more unidentified peaks but with a very similar structure to DA and specially to Iso-A. These profiles are the first reported for *P. multiseries* in Argentina. Our results are comparable to *P. multiseries* from other geographical areas such as New Zealand which also contained DA as the primary analogue and all isomers found in the Buenos Aires Province strains and additionally Iso-C, in similar low concentrations [93] (Table 1). It is likely that the first of the two unknown peaks found in the present study could correspond to the Iso-C found by Nishimura et al. [93], although its presence could not be confirmed due to lack of a Iso-C reference standard. Another close species to *P. multiseries* with similar toxin profile is *Nitzschia navis-varingica* Lundholm and Moestrup that presented DA, Iso-A and Iso-B [104].

It must be kept in mind that differences in DA production could be associated with culture conditions, culture growth phase at the time of analysis and presence of bacteria [105–109].

## 5. Final Considerations

The detailed characterisation of toxic strains of dinoflagellates and diatoms in the present study provided comprehensive and updated information on their morphology, molecular and toxin profiles. In order to mitigate the impacts of HABs on aquaculture, marine ecosystems and public health, it is necessary to consolidate and improve our knowledge on the representative toxic species that pose a potential risk in different geographical areas.

The target organisms herein considered produce three groups of toxins regulated by art. 275 tris, chapter VI, [110] (updated to 07-2022, Argentinean Food Code), which establishes maximum levels for each of them as a prerequisite for the marketing of bivalve and gastropod molluscs in order to safeguard public health. Therefore, the present results provide added value to the biotoxin monitoring program in marine coastal waters of the Buenos Aires Province, both from the purpose of resource management and risk evaluation, as well as for related research on harmful microalgae.

**Author Contributions:** Conceptualisation, E.A.S., J.A.T.K., F.R. and I.S.; methodology, J.A.T.K., F.R., A.E.R., P.R. and I.S.; software, validation, J.A.T.K., F.R., A.E.R. and P.R.; formal analysis, investigation, writing—original draft preparation, writing—review and editing, J.A.T.K., F.R., A.E.R., P.R., E.A.S. and I.S. All authors have read and agreed to the published version of the manuscript.

**Funding:** This research was funded by the Universidad Nacional de La Plata, grant 11/N 863 with the contribution of the Dirección Provincial de Pesca, Ministerio de Desarrollo Agrario (Provincia de Buenos Aires), and by the projects CCVIEO and DIANAS (CTM 20178-6066-R) from the Instituto Español de Oceanografía, with the contribution of the Unidad Asociada de Microalgas nocivas (CSIC IEO).

**Institutional Review Board Statement:** Not applicable.

**Data Availability Statement:** Not applicable.

**Acknowledgments:** We thank Juan Blanco, from the Centro de Investigacións Mariñas of the Xunta de Galicia, Vilanova de Arousa, Pontevedra, Spain for his generous help in the investigation of toxins. Thanks to Lic. Anabel Lamaro for the help and guidance with the construction of the Buenos Aires Province map and anonymous reviewers for helping us to improve the original manuscript.

**Conflicts of Interest:** The authors declare no conflict of interest.

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
