# Peer review of "Morphological, Phylogenetic and Toxinological Characterization of Potentially Harmful Algal Species from the Marine Coastal Waters of Buenos Aires Province (Argentina)"

_phycology, doi:10.3390/phycology3010006_

Round 1
Reviewer 1 Report
This study is well written and well structured, giving a consolidated view of the occurrence of toxic microalgae species in Argentinean waters. The methods are sound and adequate but the results are not novel. In my opinion this manuscripts reports a valuable study, I can recommend it for publication as is, but I do not see it as a priority for publication.
Author Response
We thank Reviewer 1 for the analysis of our manuscript and for the concepts expressed in his review report.
Reviewer 2 Report
The submitted manuscript Characterization of potentially harmful dinoflagellate and diatom species from Buenos Aires marine coastal waters (Argentina) (Kubis et al.) presents a phylogenetic and toxinological characterization of two dinoflagellates and one diatom species. The manuscript is largely descriptive and provides little innovation or new insights into HAB dynamics and distribution in the Argentine Sea. The molecular sequencing, morphological description of cells and toxin analysis, however, appears to be high quality information and competently done. On this basis, there could be justification for publication after major revision and modification of many technical details. First the title is very awkward; I would suggest a modification something like this: Phylogenetic and toxinological characterization of potentially harmful algal species from the coastal Argentine Sea (Buenos Aires Province). The Abstract is basically fine but there is no conclusion of the significance of the findings – “differences in..“ and “were discussed” are not acceptable to end the Abstract. Conclusion must be provided. The Introduction has two many short paragraphs that should be linked – e.g. the introductory paragraph is only one sentence. Link L44-45; L71-72. Materials and Methods are very complete and accurately described, with not major errors or omissions. The paragraph section Deposited material L323 should be moved to the end of the morphology section (i.e. after L156 rather than appear after the toxin methods. The first paragraph of the Results (L330-337) is confusing and does not actually describe results – some of this information could go in the Table and Figure legends. A new introduction to Results must be written. SEM figures are very nice, phylogenetic trees are clear and well described in the legends and the LC-MS spectra are fine except Fig. 9 is difficult to read for axis labels and details at the shown magnification! The Discussion is generally well done for a mainly descriptive paper but again there are too many little short paragraphs – some of these could be combined, e.g. L624-625-629-630 --- this section is all about Alexandrium. The section descriptors “morphological”, “molecular”, “toxinological” etc. must be replaced with a more complete phrase to indicate what is being discussed in each section, e.g. Comparison of morphological variation, etc. Throughout the Discussion, the authors often repeat specific details of data that belong in the Results, e.g. L779 and then we are distracted by the detail and lose the general interpretation – less numbers, please! The comparative literature is quite well cited in the Discussion but there is little evidence that these authors have done much interpretation of what other workers have found. For example, L761-763 cites previous Argentine studies for a possible explanation of the variation in toxin profile, but the “explanation” is vague and does not indicate what THESE authors think about the issue. The “Final considerations” section is not really useful and should be deleted, redistributed or replaced completely. In the current section there is a mixture of “conclusions”, “highlights”, “results” and “background information” for the Introduction. Instead this paper needs a Conclusions section which indicates why this work is important and significant and what it has to do with toxin and HAB monitoring. Finally, a clear statement of what is new and changes our perspective on what we knew before this study about the phylogeography of these HAB species in the Argentine Sea must be included.
This submitted paper is full of minor technical errors and should be proof-read by a native English reader before re-submission. A selected few of these easily corrected technical issues are indicated below.
Details: L23- replace “were studied using” with “were determined by”; number of significant figures reported in the manuscript, e.g., L31, L32 12.38 – 46.40 pg – this level of precision is not believable and is likely a calculation artifact; abbreviations for the toxins should be consistent, e.g. yessotoxin (as YTX) and domoic (as DA) are not abbreviated in the Abstract but gonyautoxin 2,3 are given here as GTX2,3; L74 missing: spirolide (SPX) producing. “DA poisoning” L89, L93 is not very commonly so described – better just mention as ASP. In M&M: “sulfocarbamoyl” is misspelled L232; “Sample toxin content” – should be singular L241; “have been carried out” – replace by: “were carried out” L256; “rum” should be “run” L308. In Results: “synonymized” is not a real word; perhaps change to “synonymous” L404; “in front of” is weird, perhaps replace with “adjacent to” L429; “early” should be “earlier” L462. Discussion: missing: “the” China Sea L650
